# Usability of the REHOME Solution for the Telerehabilitation in Neurological Diseases: Preliminary Results on Motor and Cognitive Platforms

**DOI:** 10.3390/s22239467

**Published:** 2022-12-04

**Authors:** Claudia Ferraris, Irene Ronga, Roberto Pratola, Guido Coppo, Tea Bosso, Sara Falco, Gianluca Amprimo, Giuseppe Pettiti, Simone Lo Priore, Lorenzo Priano, Alessandro Mauro, Debora Desideri

**Affiliations:** 1Institute of Electronics, Computer and Telecommunication Engineering, National Research Council, 10129 Turin, Italy; 2BraIn Plasticity and Behaviour Changes Research Group, Department of Psychology, University of Turin, 10124 Turin, Italy; 3Engineering Ingegneria Informatica S.p.A., 00144 Rome, Italy; 4Synarea Consultants s.r.l., 10153 Turin, Italy; 5Geriatrics Unit, Città della Salute e della Scienza Hospital, 10126 Turin, Italy; 6Clinical Pyschology Unit, Città della Salute e della Scienza Hospital, 10126 Turin, Italy; 7Department of Control and Computer Engineering, Politecnico di Torino, 10129 Turin, Italy; 8Istituto Auxologico Italiano, IRCCS, Department of Neurology and Neurorehabilitation, S. Giuseppe Hospital, 20123 Milan, Italy; 9Department of Neurosciences, University of Turin, 10126 Turin, Italy

**Keywords:** telemedicine, telerehabilitation, remote monitoring, health care platform, motor and cognitive rehabilitation, sleep disorders, neurological diseases

## Abstract

The progressive aging of the population and the consequent growth of individuals with neurological diseases and related chronic disabilities, will lead to a general increase in the costs and resources needed to ensure treatment and care services. In this scenario, telemedicine and e-health solutions, including remote monitoring and rehabilitation, are attracting increasing interest as tools to ensure the sustainability of the healthcare system or, at least, to support the burden for health care facilities. Technological advances in recent decades have fostered the development of dedicated and innovative Information and Communication Technology (ICT) based solutions, with the aim of complementing traditional care and treatment services through telemedicine applications that support new patient and disease management strategies. This is the background for the REHOME project, whose technological solution, presented in this paper, integrates innovative methodologies and devices for remote monitoring and rehabilitation of cognitive, motor, and sleep disorders associated with neurological diseases. One of the primary goals of the project is to meet the needs of patients and clinicians, by ensuring continuity of treatment from healthcare facilities to the patient’s home. To this end, it is important to ensure the usability of the solution by elderly and pathological individuals. Preliminary results of usability and user experience questionnaires on 70 subjects recruited in three experimental trials are presented here.

## 1. Introduction

In recent years we have been witnessing a gradual increase in life expectancy, due to the general improvement in lifestyles and advances in medicine. This phenomenon has led to the progressive aging of the population, with significant consequences for future economic and social policies [1]. Reports on aging by the World Health Organization (WHO) confirm this trend, further indicating that the population over 65 will double and the population over 80 will triple by 2050 [2]. In addition, several studies and reports on global health have highlighted that aging is inherently accompanied by an increase in the number of people with age-related diseases and disabilities [3,4] that need specific health care treatments for prolonged periods. The same reports also show that neurological diseases and acute events such as stroke, which have a prevalence in the older population, are already very common in industrialized countries characterized by greater well-being. Neurological and neurodegenerative diseases, such as dementia and Parkinson’s disease, have a dramatic impact on quality of life since they progressively induce severe and chronic disabilities in the cognitive and motor domains. Stroke is a major cause of comorbidity and disability [5], in which hemiplegia (or hemiparesis) and impaired gait are common consequences of the loss of brain function in cortical motor areas. Parkinson’s disease is recognized as the second most common neurodegenerative disorder after Alzheimer’s disease, and it causes a progressive impairment in motor functions (bradykinesia). In addition, sleep disorders are common comorbidities in the neurological and post-stroke clinical picture that require complex instrumental investigations and specific treatment to avoid consequences in daily activities [6].

This scenario thus imposes significant efforts and challenges for prolonged patient care, and in particular, rehabilitation programs that aim to mitigate the adverse effects and still ensure the best quality of life [7,8,9]. Several studies have also shown that rehabilitation outcomes are better when patients can continue rehabilitation treatment at home [10]. Moreover, it is clear that the health sector will be one of the most affected by the demographic change in the coming years, in terms of the sustainability of health services [11].

To address future challenges in healthcare, solutions based on Information and Communication Technology (ICT) [12] are attracting increasing interest in finding a trade-off between patient needs (quality of life), effectiveness of healthcare services (monitoring and rehabilitation protocols), and sustainability of the healthcare system (cost and resources). For example, ICT solutions could support new patient and disease management strategies based on telemedicine and related applications, thereby moving monitoring and rehabilitation services from health care facilities to home settings [13,14], thus favoring continuous and remote follow-up of patients. However, there are still several barriers that limit its deployment in the home environment, including aspects of usability, acceptability, lack of motivation and skepticism in using digital tools, especially in the elderly and individuals with severe chronic disabilities [10,15,16].

This is the background for the REHOME project [17], whose technological solution is presented in this paper. The project involved twelve partners (including seven small and medium-sized enterprises, three research institutions, and two hospitals) and was developed in a multidisciplinary context of technological and clinical expertise to achieve a helpful, comprehensive, and integrated home solution. The project focuses on the remote monitoring and rehabilitation of cognitive, motor, and sleep disorders originating from neurological diseases and injuries, in particular stroke, mild cognitive impairment, and Parkinson’s disease. To this end, the proposed solution integrates innovative technologies and methodologies able to ensure patient engagement and continuity of care, monitoring, and rehabilitation in supervised and minimally supervised scenarios. The REHOME solution exploits different types of sensors and devices (such as optical sensors, electromyography, commercial and prototypal sensors for physiological signals); innovative methods for rehabilitation (such as virtual reality, exergaming, gamification techniques); traditional and gamified motor tasks, derived from standardized clinical scales, to evaluate patients’ current condition; estimation of objective features to quantify the patients’ performance and progression over time. At the same time, it facilitates communication and interaction between doctor, patient, and caregiver through the infrastructure of the healthcare platform and its web-based facilities.

In recent years, home-based rehabilitation has been widely considered and many solutions have been developed for this purpose, as evidenced by some recent literature reviews. The study by Hosseiniravandi et al. [18] compared twenty-two solutions for remote rehabilitation focused primarily on five categories of diseases and disorders, including musculoskeletal, neurological, respiratory, cardiovascular, and other general health-related problems. The analysis revealed that these solutions shared three common functionalities, namely exercise plan management, outcome reporting, and patient education. There were also many similarities in terms of methods to collect data: automatic data collection (85%), recording of patient treatment progress (90%), and providing periodic (75%) and real-time (85%) feedback to therapists and patients. In contrast, the study did not focus on the hardware components and devices used in the solutions analyzed, nor on the real-world implemented systems, infrastructure, and the effects of these systems in real world settings. The study by de Souza et al. [19] examined fourteen home-based rehabilitation frameworks based on gaming or exergaming approaches. The findings of the study showed that most of the solutions (79%) focused on stroke, motor, and cognitive rehabilitation. In addition, only 22% of them were cloud-based applications. In [20], a detailed overview of telerehabilitation and its fields of application was provided, with an analysis of the benefits and drawbacks associated with its use. The study highlighted the main disadvantages of telerehabilitation, including patient skepticism due to remote interaction with therapists. The study also suggested that further research is needed to improve electronic equipment and devices, and to make applications as flexible as possible to increase the reliability and effectiveness of telerehabilitation equipment for treating patients with specific problems. The effectiveness of telerehabilitation was investigated in post-stroke subjects and individuals with mild cognitive impairment in [21,22], respectively. Both studies concluded that telerehabilitation can be an appropriate alternative to the usual rehabilitation care.

However, as also pointed out by the previously mentioned state-of-the-art studies, several challenges remain [15,23]. For example, patient satisfaction, patient involvement, and acceptability of the proposed remote rehabilitation approach are generally given little consideration. The same occurs for physical discomfort caused by sensors and devices, which are often invasive or complex to use. In most cases, a single methodological approach is provided that is not suitable for different therapeutic areas, pathological conditions, usage scenario, and current patient status. In addition, a proper user interface design, which is critical especially in the case of people with disabilities, is often neglected. Finally, the lack of adequate training on the use of technological devices and solutions is another common weakness that often prevents patients from using them effectively and continuing the planned therapeutic treatment.

The REHOME project, and the implemented solution, addressed some of the weaknesses highlighted, such as increasing patient involvement; understanding and addressing the specific and necessary functions for the different target pathologies (each with its own peculiarities); managing the personalized treatment plan and evaluating the effectiveness of remote treatment; addressing aspects related to usability, taking care of the user interface, interaction methods, and the simplicity of the devices involved; integrating multiple sensors and methodological approaches to evaluate different domains (motor, cognitive, and sleep) but common to the target pathologies. All these features represent the innovative and peculiar aspect of the proposed solution with respect to the state-of-the-art.

Currently, the project is in its final experimental phase at hospitals. However, some pilot trials were previously organized with the aim of obtaining feedback on the usability of some tools included in the implemented solution. The main objective of the paper is to present the preliminary results related to usability, one of the key elements is to propose a suitable and helpful solution for remote monitoring and rehabilitation. Specifically, the contributions of this work concern the following points:Introducing the technological solution developed in the REHOME project, highlighting its main components, innovative features, and methodological approaches to meet the needs of patients and healthcare professionals and overcome the main weaknesses of telemedicine and eHealth services that emerge in the literature;Presenting three experimental protocols concerning the motor and cognitive platforms that involved groups of elderly patients affected by Parkinson’s disease (and forms of atypical parkinsonism) and mild cognitive impairment, target pathologies of the REHOME project;Presenting the preliminary results on the usability and user experience evaluation using questionnaires administered to the participants to get feedback on the strengths and weaknesses of the developed platforms.

Indeed, it should be noted that this paper is an extended and more detailed version of the one recently presented at the second IEEE Conference on ICT solutions for eHealth (ICTS4eHealth 2022) [24].

The next sections are organized as follows: Section 2 presents the implemented solution, describing the overall architecture and its main components, with a focus on those considered for usability evaluation; Section 3 presents the main results on the usability questionnaires according to the organized pilot trials; Section 4 contains discussion and future developments; Section 5 contains some concluding remarks.

## 2. Materials and Methods

### 2.1. The Architecture of the Solution

REHOME is a tele-rehabilitation system for personalized and gamified patient training and monitoring by Healthcare Professionals (HCPs). Patients at home can carry out exercises according to their own rehabilitation plan by using the enabling device kit provided by the doctor. Rehabilitation and health evaluation session data are then collected to allow remote and timely analysis and, if deemed necessary, improvements to the plan.

As depicted in Figure 1, the REHOME system, developed with cloud technologies and based on a distributed microservices architecture, is composed of several subsystems:HCP Platform (HCPP): to monitor patients remotely and to assess their progress;Cognitive Rehabilitation and Gaming Platform (CRGP): based on gaming to train five different cognitive domains and to improve memory and orientation skills;Motor Rehabilitation and Exergames Platform (MREP): for automatic assessment and rehabilitation of motor disabilities concerning limbs, posture, balance, and coordination;Sleep Evaluation Platform (SEP): to detect and evaluate sleep disorders.

**Figure 1 sensors-22-09467-f001:**
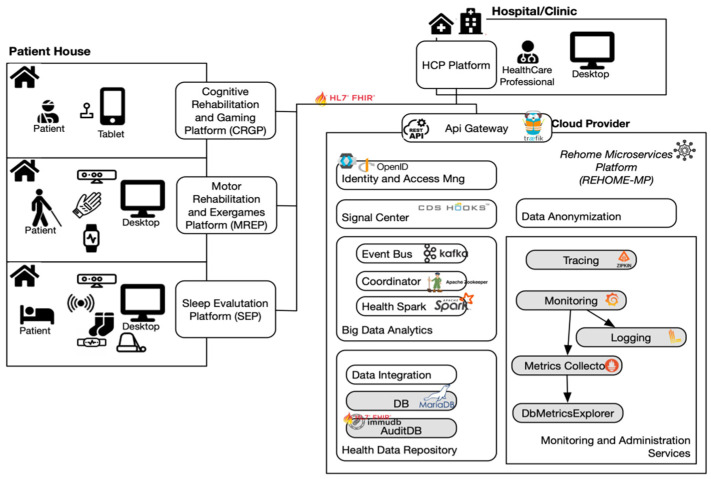
REHOME system architecture.

Motor, cognitive, and sleep platforms are located at the patient’s home and rely on different kinds of devices/sensors. On the other hand, the HCP platform is located in hospitals and clinics.

Finally, the REHOME Microservices Platform (REHOME-MP), developed by Engineering Ingegneria Informatica S.p.A. (Rome, Italy), is deployed in the cloud and provides the core functionalities of the system. Such a platform consists of several components, designed according to the microservices principles to be extendable and support new future needs of telemedicine-related platforms. Microservices are self-contained applications that can be developed and deployed individually, but that work together to deliver an overall solution. The microservice architecture makes it possible to achieve full independence of each individual component that can be modified, without affecting other services or functional aspects of the system.

Each component of the platform was introduced to meet clinical and technical requirements. The requirements specification process leveraged the scenarios and personas to define the main actors and, for each one, the most relevant expected interactions with REHOME. After that, the analysis of the scenarios gave rise to set use cases, formalized according to the model proposed by [25], and accordingly user and system requirements.

Among the requirements, particular emphasis has been provided on the interoperability. Interoperability among the different platforms is, indeed, guaranteed by an API Gateway based on the HL7 FHIR R4 standard [26] for the exchange of health data, which supports RESTful communication over a HTTP/s protocol, OpenID Connect security standards and encrypted channel. The whole architecture is designed according to an event-driven pattern to decouple supporting microservices from disease-specific platforms [27]; this approach enhances scalability and integration of additional platforms. The core element enabling this pattern is the Event Bus (based on [28]), designed to collect and dispatch data according to the HL7 FHIR.

The interactions with the Event Bus are mediated by the API Gateway, which is responsible for dispatching messages to the various microservices, enforcing all the security-related policies.

The main microservices of the REHOME-MP are described hereafter.

The Health Data Repository collects, integrates, and shares data coming from the different platforms. The data have been harmonized according to the HL7 FHIR data model. The approach, used to integrate different data consists of (1) defining a conceptual model, represented by UML class diagrams, (2) mapping each class on the HL7 FHIR data model by leveraging the LOINC common dictionary. Data in the health repository are managed in compliance with the GDPR requirements so as to ensure the safety of the personal data of the patients. In that sense, basic identity information and health data are managed in separate data stores protected by ensuring (among others), encryption at rest and in transit. Such data are partitioned and minimized to ensure that HCP can access only relevant and authorized information.

The Big Data Analytics provides services for analyzing the amount of heterogeneous data collected by the platform such as: clinical rehabilitation data, continuously monitored data like bio-potential signals coming from the MREP and SEP platform sensors, multimedia contents like video recording of rehabilitation and sleep sessions. The system deals with managing both real-time analyses (data are processed as soon as they are available) and batch analyses (blocks of data are processed at specific time intervals). This service is used to produce new knowledge, starting from the monitored data, that will be used by the system to both infer information and produce notifications and collaborating with the Signal Center, to support the HCPs (Healthcare Professionals).

The purpose of the Signal Center is indeed to notify end-users, about events of interest prior to a subscription procedure. Examples of supported notifications are: “the patient skipped one or more training sessions”, “the score of some exercises is too low”, “time to perform a training session too long”, and so on. The system is based on the HL7 CDS-Hooks, an open-source specification focused on triggering robust clinical decision support [29] in electronic health records.

The Signal Center APIs allow the creation of subscriptions for receiving notifications on different communication channels; each subscription is associated with rules written using HL7 CQL, a language specialized for the clinical domain [30]. The framework processes the data as soon as they arrive and if the rules are verified, a notification is generated as a CDS-Hooks card (informative cards, presenting recommendations or suggestions). The data sources for the CQL rules are the Health Data Repository and the new knowledge generated by Big Data Analytics.

The data anonymization produces anonymized datasets starting from data of a FHIR-compliant repository (such as the Health Data Repository). The service is configurable and extendable with different anonymization algorithms. The component follows an Extract, Transform and Load (ETL) process which involves: (1) the extraction of FHIR resources from the Health Data Repository, (2) filtering of relevant attributes, (3) anonymization of the extracted resources [31,32]. Those operations allow the HCPs to select the anonymization properties to be applied on specific attributes, and eventually, export data in databases or CSV files. The supported anonymization policies are k-anonymity and i-diversity.

Anonymization and in general, security requirements are met to make the platform compliant with Regulation (EU) no. 2016/679 (also known as GDPR). In that sense, the platform can manage the following security aspects: authentications and authorizations by mean of the Identity and Access Management component, compliant to the OpenID Connect (OIDC) protocol (OpenID Foundation, San Ramon, CA, USA) and based on an open source identity provider Keycloak (WildFly, Raleigh, NC, USA) [33]; auditing to track operations on data and store them in an immutable audit database (based on ImmuDB (Codenotary Inc., Bellaire, TX, USA) [34]); and data encryption at rest and in transit.

Finally, REHOME-MP also introduces infrastructural services for monitoring purposes, necessary due to the distributed nature of microservices architecture, such as: tracing service to observe call sequences among microservices (based on open source Zipkin (Twitter, Inc., San Francisco, CA, USA) [35]); monitoring dashboard to view real-time infrastructure metrics (based on open source Grafana (Grafana Labs, New York, NY, USA) [36]) collected by a centralized module (based on open source Prometheus (Linux Foundation, San Francisco, CA, USA) [37]) and centralized logging (based on the Grafana Loki stack [38]).

The main advantages of the proposed architecture come from the adoption of standards, architectural patterns and open-source technology. As already highlighted the system is flexible and can be easily extended by integrating additional disease specific monitoring platforms. For these reasons, the architecture design can be applied to other healthcare-related contexts.

### 2.2. HCP Platform

The HCPP (Engineering Ingegneria Informatica S.p.A. (Rome, Italy) allows management of patient data and personalized rehabilitation plans for the supported diseases (e.g., scheduling of evaluation sessions or exercises with personalized settings). The HCPP consists of a fat-client web application developed by leveraging the open source Angular framework (Google LLC., Mountain View, CA, USA). It also provides support for visualizing monitored data and notification mechanisms for evaluating the rehabilitation’s sessions through:generic dashboards, common to the various diseases, summarizing the salient data in graphs and diagrams like scores, times, adherence, statistics, and variations over time of variables of interest;specific dashboards for each disease, tailored according to the needs and requirements of each platform (like views for specific data or domain specifics graphs, including video recording of training sessions).

The data represented in the HCPP are retrieved from REHOME-MP by leveraging the capabilities of Big Data Analytics while aggregating and analyzing data to facilitate their representation to HCPs and provide clinically relevant information. In addition, HCPP supports HCPs in defining and personalizing treatment plans. To this end, HCPs can: select the type of monitoring and rehabilitation treatment (motor, cognitive, sleep); assign rehabilitation and assessment tasks, functions, and exercises to the treatment plan; configure parameters for tasks and exercises, such as the game level difficulty; define the plan duration, days to carry out the sessions, and optionally, the maximum duration of each session.

Figure 2 shows an example of generic and specific dashboard related to monitoring of sleep disorders, creation of the personalized rehabilitation plan, overall summary of sessions performed by the patient.

The generic dashboard shown in Figure 2a summarizes the performance monitored through the dedicated platforms (MREP, CRGP, and SEP) to provide a comprehensive overview of the patient condition to HCPs. Figure 2b is the specific dashboard for CRGP that presents information related to various cognitive domains, including scores, trends, and time information. Figure 2c is an example of the specific dashboard for MREP: on the left, the kinematic parameters and the automatic evaluation of a specific motor task (in this case, leg agility) are shown using radar plots and tables; on the right, the signals and gesture classification, collected through an electromyography armband during upper limb rehabilitation, are shown. Figure 2d shows data collected by SEP, including physiological signals, environmental parameters, and video recordings of anomalous events detected during the sleep session.

### 2.3. Cognitive Rehabilitation and Gaming Platform

The Cognitive Rehabilitation and Gaming Platform (CRGP) focuses on training and rehabilitation of specific cognitive functions/domains that are typically compromised in older people with mild neurocognitive disorder (DSM-5), due to different etiologies such as Alzheimer’s disease (AD), frontotemporal lobar degeneration, Lewy body disease, vascular disease, traumatic brain injury, substance/medication use, HIV infection, prion disease, Parkinson’s disease, Huntington’s disease, “another medical condition”, multiple etiologies, and unspecified.

The CRGP is accessible via a mobile application for tablets and provides exercises from three main domains: (i) single domain and ecological exercises, (ii) spatial memory domain, and (iii) cognitive–motor rehabilitation domain. According to a tailor-made clinical approach, all the game-exercises are designed to train different cognitive functions (such as attention, memory, and executive functions), in order to tend the multiple-rehabilitative needs of patients affected by cognitive impairments. Elements of gamification are provided for each domain, in order to enhance patients’ engagement and intrinsic motivation, thus encouraging constant exercise and commitment in the long term [39]. A summary of the main functionalities provided by the CRGP is shown in Table 1.

Specifically, single-domain exercises were designed to train specific cognitive functions, including attention, executive functions, long-term memory, and learning, through the rationale of repetition, gradualness of difficulty, and customization to the patient’s current improvements and abilities [40]. A gamified approach and a 360-degree exploration of the environment, which recreates typical contexts of daily life, ensured patient involvement and motivation [41]. The ecological exercises, on the other hand, were designed to reproduce habitual situations of everyday life [42], thus enabling contextualization of cognitive function training to everyday life skills. The suite of single-domain and ecological exercises were developed by Integrated Solution s.r.l. (Turin, Italy), a technical partner of the REHOME project.

The cognitive–motor rehabilitation domain, on the other hand, involves training of cognitive–motor skills, including motor control and motor inhibition. To this end, a 3D videogame was designed. The game consists of the exploration of a desert island, with the aim of locating different places (such as the beach or a waterfall), where specific mini-games, of progressively increasing difficulty, are activated. Mini-games were designed to train and improve topographical orientation, object categorization, motor planning, learning and recall of visual information, and motor inhibition (as a go-nogo task [43,44]). The 3D videogame was designed by BIP Research Group (University of Turin) and developed by Black Flamingo s.r.l. (Turin, Italy) for graphics and SynArea Consultants s.r.l. (Turin, Italy), a technical partner of the REHOME project, for 3D virtual game and data collection.

Regarding the spatial memory domain, since it was the focus of the usability evaluation (CRGP platform), a more detailed description has been included in Section 2.3.1.

Since ICT solutions ensure remote monitoring, CRGP allows directly personalizing rehabilitative plans based on feedbacks provided by the system itself and by the patient’s report, thus further increasing the level of adherence to the training [45].

#### 2.3.1. Spatial Memory Domain

The spatial memory domain was trained by using a specific virtual navigation task, capable of stimulating the ability to form spatial allocentric representations from the first-person perspective navigation [46]. This is based on the idea that visuo-spatial skills are able to support the maintenance of other cognitive abilities, such as memory, mental imagery, and reasoning [47,48]. Since virtual reality (VR) has proven to be an effective tool for cognitive rehabilitation [49], an interactive 3D game for spatial navigation, *MindTheCity!*, was developed, thus providing patients with a more ecological, safer and engaging activity. The virtual environment represents an imaginary city organized into five districts with progressively increasing dimensions (and therefore progressively increasing navigation difficulty). The training goal is to find and collect five components of a bicycle (i.e., wheels, handlebars, pedals, rear frame and front frame) dislocated in the districts. Buildings and streets in the city are all alike and few environmental landmarks are available (e.g., a central garden—which is always the starting point for the exploration-, a church, a skyscraper, a telecommunication tower), in order to make orientation more difficult and stimulate the use of survey strategies and the creation of cognitive maps of the environment in allocentric frameworks [50,51]. Notably, the virtual environment can be explored with only the support of visual information, since auditory cues are intentionally excluded from the game. Specifically, the wayfinding task is divided into two phases: (i) in the first phase (*free exploration*), participants are asked to explore the neighborhood freely in order to locate all bike components without collecting them, (ii) in the second phase (*object search task*), participants are asked to retrieve the same objects by following the shortest path. In this phase, whenever the user finds an object (Figure 3a), they are asked to perform a *pointing task* through her/his avatar (Figure 3b), pointing an arrow toward the starting point (for the first object) or toward the previously retrieved bike component (from the second object on). The pointing task offered the degrees of angular error between the avatar’s pointing and the targets’ actual positions, thus providing a behavioral measure of patient’s success in representing the environment in allocentric coordinates.

The 3D game, whose graphics and the software for data collection were realized by SynArea Consultants s.r.l. (Turin, Italy), was administered through portable devices (tablets), to be used in domestic settings. Users’ interaction was allowed through a joystick. The interactive game has been developed in Unity^®^, adopting a low-poly graphics mode to avoid lags that could interfere with and disrupt user performance, and it is delivered both on mobile devices (tablets) and PCs to also be used in domestic contexts. For the experimental trial, a tablet Samsung Galaxy Tab S5e with the following hardware configuration was used: Opta-Core processor (2.0 GHz), 4 GB of RAM, 64 GB of ROM, 10.5 inch display, LTE connection, and Android operating system.

The game is configurable according to the patient’s condition through specific parameters and tips to facilitate the tasks. Configuration parameters include the setting of the exploration difficulty level of city districts; availability of a 2D mini-map that shows the position of the avatar and objects inside the city; use of the virtual joystick; visualization of the objects through the walls of buildings; activation of arrows that suggests the position of the nearest object.

Results of users’ virtual space exploration, as well as users’ abilities to retrieve the bicycle components, were recorded and sent to the REHOME-MP and shared with HCPs for the remote patient’s monitoring. Results of each game session are sent to REHOME-MP using JSON format data structures and Fast Health Interoperability Resources (FHIR) protocol procedures.

Figure 3a shows the avatar exploring the city to retrieve the bicycle components (search task). The bike components to be found are displayed in red. The distance is shown at the top center of the scene, and the objects already found are in the upper right corner. When an object is found, the patient must select it to collect it. Then, the object turns green, indicating that it was correctly collected. The results of the game session are shown in Figure 3c, which displays general and specific information, including the difficulty level, number of objects found, time information, exploration path, and the final score.

### 2.4. Motor Rehabilitation and Exergames Platform

The Motor Rehabilitation and Exergames Platform (MREP) addresses the assessment and rehabilitation of movement disorders of Parkinson’s disease and post-stroke, but it can also be used to evaluate motor decline in the elderly with cognitive impairment. MREP provides both rehabilitative exercises and evaluative motor tasks related to traditional clinical assessment scales, including Unified Parkinson’s Disease Rating Scale (UPDRS) and Motor Evaluation Scale for Upper Extremity in Stroke (MESUPES) [52,53]. This aspect is particularly important from the clinical perspective: the platform, using motor evaluations based on validated scales, also integrated with the performances during exercises, is therefore able to provide useful information to the clinicians in the way they are accustomed to. This information can then be used to constantly tailor medical treatments and rehabilitation.

To achieve its end, MREP employs user-friendly sensors (i.e., RGB-Depth sensor and armband based on surface electromyography) and innovative methodologies (including virtual reality, gamification, non-invasive body tracking) to be easily usable even in an unsupervised scenario, such as a patient’s home.

MREP communicates with the REHOME Microservices Platform (REHOME-MP) through an HTTP-based protocol to get the patient’s personalized treatment plan and to send specific data and parameters related to the patient’s motor performance collected through sensors and analysis procedures. In fact, evaluative and rehabilitative exercises are configured and assigned properly according to the patient’s condition and rehabilitation needs, thus implementing a tailor-made treatment plan. MREP addresses three main areas: (i) assessment of the motor function through optical approaches, (ii) rehabilitation and training of upper limb and trunk movements through exergames, (iii) rehabilitation of arm and hand functions through surface electromyography (sEMG). A summary of the main functionalities provided by the MREP is shown in Table 2.

Rehabilitation of upper limb and hand motor function is based on REMO^®^ (Morecognition s.r.l., Turin, Italy), a sEMG-based armband that detects forearm muscle activation ([54,55]) during the execution of typical movements used in clinical practice (such as finger extension, cylindrical grasp, and spherical grasp). A standalone application was also developed to simplify device management and muscle activation monitoring through an ad-hoc GUI that provides a visual feedback of electromyography signals and counts correct movements [56]. The sEMG signals and performance results are sent to the REHOME-MP at the end of each acquisition session for analysis by HCPs.

Regarding the other two components of the MREP platform, a more detailed description has been included in the next subsection since they were evaluated for usability.

The availability of data and parameters, estimated by MREP components, on the HCPP allows HCPs to remotely monitor any improvement or worsening of the motor condition, thus intervening promptly in case of unexpected motor behavior or to stimulate new rehabilitation goals.

#### 2.4.1. Assessment of the Motor Function through RGB-Depth Sensors

This component of the MREP platform implements a vision system, built around a single RGB-Depth sensor, to capture human body movements in real time. In particular, the vision system is based on the new Microsoft^®^ Azure Kinect DK camera (Microsoft Corporation, Redmond, WA, USA) [57] and its body tracking algorithm [58]. It also includes a high-performance processing unit (e.g., laptop or mini-computer) and a monitor (or TV screen). Specifically, a laptop with the following hardware configuration was used: 9th generation Intel^®^ Core^TM^ processor (2.4 GHz quad-core), 16 GB of RAM, NVIDIA GeForce RTX 2060 6GB GDDR6, Windows 10 operating system. With regard to the Azure Kinect sensor, its performance has already been analyzed and validated against gold-standard systems in several recent studies [59,60,61], denoting superior accuracy, tracking robustness, and 3D reconstruction of human body movements compared to both previous device models and other commercial optical devices [62,63,64]. In the last years, RGB-Depth sensors have been widely used for the analysis and evaluation of specific motor disorders and disabilities [65,66,67,68,69,70,71,72]. The same is happening with Azure Kinect, which is beginning to be used in clinical trials [73,74,75].

The system uses two different tracking algorithms aimed at capturing coarse body and fine hand movements. The body tracking algorithm of Azure Kinect is used for real-time capture of body motion through a 3D skeletal model consisting of 32 joints that correspond to anthropometric points on the body, as in [76]. To capture fine movements of the bare hand, a 3D version of Google Mediapipe Hands [77,78] has been implemented by exploiting the depth information provided by the camera, as described in [79]: in this way, an accurate 3D skeletal model of the hands is also available. The 3D skeletal models are mainly used to characterize motor condition through a series of evaluative tasks and exercises to objectively measure specific functional parameters. Specifically, the following categories of exercises have been included:Body motor tasks: traditional evaluative tasks derived from the UPDRS and balance scales, including leg agility (LA), sit-to-stand (S2S), gait (G), posture and postural stability (PoS), suitable for both parkinsonian and post-stroke hemiplegic subjects which belong to this category.Upper limb motor tasks: traditional evaluative tasks for the upper limbs derived from the UPDRS (i.e., finger tapping, hand movements, pronation and supination) and the MESUPES-Hand scale [53] (specifically, a subset of range-of-motion tasks) which belong to this category.Motor tasks in the virtual environment: to this category belong two exergames that stress motor control and coordination, specifically Lateral Weight Lifting (LWL) and Frontal Weight Lifting (FWL) of the upper limbs, and the exergame Bouncing Ball (BB), a gamified version of traditional leg agility.

The analysis procedure, consisting of custom-written MATLAB^®^ (Mathworks Inc., Natick, MA, USA) scripts, works on the collected trajectories of the 3D skeletal models to perform the movement analysis by estimating objective functional parameters and detecting anomalies to evaluate the overall motor condition.

To ensure high usability of the system, which is particularly critical in the case of elderly people with disabilities and normally poor technological skills [16], special attention has been paid to human–machine interaction (HMI) and graphical user interface (GUI) design to facilitate self-management of the system during the data acquisition phase, even in minimally supervised settings (for example the subject’s home).

The HMI is based on the same tracking algorithms and 3D skeletal models used to assess motor conditions, thus ensuring a natural, contactless, noninvasive, and more intuitive interaction during the data acquisition. Second, the GUI consists of Unity^®^ scenes (Unity Technologies, San Francisco, CA, USA) [80] in which a few interactive objects (e.g., buttons) are organized so they can be reached with simple hand movements. Interactive objects are also appropriately resized to be visible in case of reduced sight, typical of elderly people. In addition, the system automatically rearranges GUI objects according to the healthiest body side set in the system configuration file, to prevent excessive movements in case of reduced mobility. Specific configuration parameters are available for motor tasks in virtual environments, including difficulty level, number of movements to be performed, and exercise duration. It is important to note that settings may be different for the right and left sides of the body, thus taking into account a different impairment severity.

Text and voice messages, thanks to the integration of text to speech functionality, complement the GUI to guide the user in performing the assigned traditional body motor tasks (Figure 4a), motor tasks in a virtual environment (Figure 4b), and upper limb motor tasks (Figure 4c).

Figure 4a shows the scene displayed for traditional evaluative tasks. The scene includes text messages guiding the patient to perform the task, the “STOP” button to interrupt the execution, and the time remaining to complete the task. The red box shows the patient’s body captured in real-time by the Azure Kinect. A similar layout is shown in Figure 4b for evaluative motor tasks in a virtual environment. In this case, the scenario is the gymnasium. Text messages for the patient are displayed in the upper left corner. Points (correct movements) and errors (incorrect movements) are also displayed in the scene. Figure 4c shows the landmarks of the hand skeletal model captured by Google MediaPipe during the upper limb tasks for PD, namely finger tapping, hand movements, and pronation–supination of the hand.

In addition, the system is designed to manage many operations automatically, including authentication of the procedure and downloading of the personalized rehabilitation plan, assigned by the HCPs to assess motor condition jointly with rehabilitation exercises, from REHOME-MP, execution and analysis of motor tasks, data transmission to REHOME-MP. Data exchange with the REHOME-MP occurs using JSON format data structures and FHIR protocol procedures.

#### 2.4.2. Motor Rehabilitation with Exergames in a Virtual Environment

This component of the MREP platform includes 3D exergames in virtual environments designed and developed for motor rehabilitation according to the clinical and therapeutic specifications of the HCPs. In recent decades, the potential benefits of virtual reality [81,82,83,84] and exergaming [85,86,87] have been extensively investigated for rehabilitation purposes in several pathologies, as an innovative tool to stimulate motor and cognitive functions, even simultaneously, highlighting the advantages in joining them to traditional therapeutic protocols [88]. The primary role of exergames is to promote specific motor and cognitive tasks through the experience of playing games in a virtual environment, with the goal of increasing engagement through positive feedback and gamification features typical of video games (motivation, rewards, fun) and, at the same time, to encourage stimulating new rehabilitation goals in the medium–long term as planned by HCPs [83,89,90].

This component uses the same vision system, based on Azure Kinect and human body tracking facilities, and the same configuration defined for the assessment of motor condition, as described in Section 2.4.1. In this way, the 3D exergames, developed for rehabilitation purposes, integrate with the assessment aspects to be activated according to the personalized treatment plan assigned by the HCPs to assess the motor condition along with the rehabilitation exercises. Exergames have been developed in Unity^®^, adopting a low-poly graphics mode to ensure real-time capture of body movements without lags that could have interfered and disrupted user performance. Interaction with the virtual environment was through specific body movements, particularly challenging for subjects with neurological issues, defined for each exergame by HCPs. The 3D skeletal model, captured with the Azure Kinect, was mapped onto the game avatars in the virtual scene. Specifically, the following exergames have been developed and integrated:Cross-country skiing (CCS): this exergame has been designed to stimulate synchronized and alternating movement of the upper limbs. Continuous and rhythmic movements of the upper limbs make a virtual skier (avatar) move on a cross-country track to the finish line. When the movement of the upper limbs is interrupted or is not rhythmic, the skier stops, thus stimulating the patient to resume the correct movement (Figure 5a). The scene reproduces a snowy scenario. Several gems are placed on the track, stimulating the patient to collect them by moving the avatar correctly. Total points and elapsed time are displayed in the upper right corner.Airplane (PLANE): this exergame has been designed to stimulate trunk movements and upper limb control. Trunk movements, while the arms are held in lateral extension at shoulder height, guide a virtual airplane (avatar) on a flight pathway consisting of a few rings and obstacles placed along the track. The goal is to guide the plane through the rings, avoiding the obstacles, to the end of the track. This game stimulates the user in moving the trunk correctly and continuously, while simultaneously keeping the arms in extension (Figure 5b). The scene reproduces a flight scenario. Colored circles on the flight path indicate the trajectory to follow, thus stimulating the patient to make trunk movements to pass only through the circles of the correct color, avoiding the others and obstacles. Total points and elapsed time are displayed in the upper right corner.Keyboard (KEY): this exergame has been designed to stimulate control of arm pointing and extension abilities. Arm movements move a virtual hand on a keyboard composed of colored keys, which light up in a predetermined sequence. Each key is associated with a sound. The goal is to repeat the proposed sequence by pressing the corresponding keys, while keeping the arm extended frontally, to compose a short “melody”. Pressing non-illuminated keys does not produce the associated sound, thus stimulating the user to correct the arm position (Figure 5c). The scene shows a piano with five colored keys. The keys light up one at a time in a random sequence, thus stimulating the patient to correctly point the extended arm at the lit key. Total points and elapsed time are displayed in the upper right corner.

All the exergames are performed frontally to the Azure Kinect camera and in a sitting position, thus avoiding excessive fatigue and loss of balance during the exercises. Each exergame also includes several levels of increasing difficulty. Specifically, twelve levels are available for CCS that differ in avatar speed, arm movement amplitude, track length and shape (straight path only or path with curves); nine levels are available for PLANE that differ in airplane speed, trunk movement amplitude, takeoff phase activation, rings with different colors (cognitive task); three levels are available for KEY that differ in the number of keys to press according to a random sequence and choice of the arm to be used in the rehabilitation task. The general configuration of exergames also includes the hand to be used for interacting with the game selection menu.

The customization of the exergame aims to respond appropriately to the patient’s current motor condition and progressive improvements, thus stimulating patients to improve their performance and pursue new rehabilitation goals and game scores. To this end, gamification elements have also been introduced in each exergame to adequately reward performance and ensure continuity of the rehabilitation plan. Continuity of treatments is, in fact, necessary in order not to lose the benefits gained from traditional physiotherapy treatments, by taking advantage of patients’ greater engagement and enjoyment. Examples of gamification techniques include point rewards, timed challenges, and new challenges levels (e.g., longer and more complex tracks and sequences). It is important to note that all the exergames also implicitly include cognitive aspects, particularly stimuli for the domains of attention and memory. The configurability of the exergames and the inclusion of gamification elements aims, on the one hand, to intercept the patient’s specific disabilities and needs (personalization); on the other hand, to stimulate the patient to achieve ever greater goals. All exergames were realized by SynArea Consultants s.r.l. (Turin, Italy) using the body tracking functionalities supported by Azure Kinect for user’s interaction with the game environment. This component is closely connected with the motor condition assessment component, from which it receives the treatment plan downloaded from REHOME-MP and to which it shares the results of the play sessions to be sent toward REHOME-MP. The data exchange is done through files in JSON format.

### 2.5. Sleep Evaluation Platform

The sleep evaluation platform (SEP) component aims to detect and evaluate sleep disorders, focusing on PD and post-stroke subjects [91]. SEP will mainly provide an assessment of sleep quality and possible causes of sleep disruption, but it is not aimed at supplying fine diagnoses. Nevertheless, the information regarding sleep quality is particularly important in the clinical context as sleep disorders, if not recognized, can impact diurnal performances during cognitive and motor rehabilitation. The evaluation consists of estimating sleep stages and cardiac/respiratory/movement parameters, through an automatic scoring of the polysomnography (PSG). PSG includes many signals such as electroencephalogram (EEG), electromyography (EMG), electrocardiogram (ECG), electrooculogram (EOG), pulse oximetry, respiration and Periodic Leg Movements occurring during Sleep (PLMS). The PSG makes it possible to observe sleep efficiency, sleep quality, and sleep.

To this end, the system uses a combination of non-invasive wireless sensors to assess and report sleep patterns. For standard PSG acquisition, a group of sensors has been adopted, including some off-the-shelf equipment and a set of prototypal wearable sensors (chest strap, cap, and socks) specifically developed in sensorized fabric (ASTEL s.r.l., Pavone Canavese, Turin, Italy). An environment sensor has been added to measure room noise, temperature, humidity, and illumination. The commercial pressure band Emfit QS (Emfit Ltd., Vaajakoski, Finland) supplies information on the presence in bed, the quantity of movement, and measures of heart rate and breath rate. Finally, an RGB-Depth camera, which mounts an infrared camera, has been added to provide video documentation of relevant events occurring during the night and infer more accurate information on specific limb movements [92]. A summary of the main signals and sensors provided by the SEP is shown in Table 3.

Sleep monitoring sessions are scheduled based on a personalized therapeutic plan: the modular architecture permits activation of only the sensors necessary for specific monitoring, thus minimizing patient discomfort during the night. SEP is designed to ensure high usability and flexibility in clinical and domestic settings. In both cases, a very simple web interface allows the user (specialist in clinical settings or patient in domestic settings) to manage the system with smartphones or PCs. A mini-PC completes the system: it deals with sensor functioning and management, data storage, signal processing and the transfer of processed data to the REHOME-MP.

At the end of each sleep session, the system analyses the acquired data to perform an automatic scoring [93,94], thus producing color-coded graphs representing the estimations of sleep stages (REM, NREM, awakenings), respiratory and movement events, finally assigning a summary score of sleep quality. To this end, it is possible to obtain predictive sleep staging using machine and deep learning approaches on the physiological signals collected during sleep [95,96,97,98].

All substantial data and video clips of relevant events detected during sleep are sent to the REHOME-MP for further examination and analysis through data structure in JSON format and FHIR protocol procedures.

### 2.6. Usability Evaluation for CRGP and MREP Components

This section describes the experimental protocols and questionnaires administered for usability evaluation. Data were collected during preliminary experimental trials on different groups of subjects to obtain feedback on specific components of the CRGP and MREP platforms, as shown in Table 4. The experimental campaigns were organized separately because, at the stage of project development, it was necessary to investigate different aspects solicited by the platforms on the target population. To this end, ad-hoc and standard questionnaires were administered, precisely to highlight the peculiarities of the platforms and subjects involved. The following subsections describe the experimental protocols, the subjects involved, and the objectives of the usability questionnaires.

#### 2.6.1. CRGP: Spatial Memory Domain on Elderly People with Mild Cognitive Disorders

A total of twenty-eight patients (14 males and 14 females) affected by MCI were recruited after an initial neuropsychological assessment aimed to enroll only patients who were independent in the realization of their daily-life activities. All patients performed 4 weeks (20 min per day, for five consecutive days) of experimental conditions (i.e., *MindTheCity!*), followed or preceded (the order of the experimental conditions and control was counterbalanced across patients) by another 4 weeks of control condition (i.e., passive observation of videos of *MindTheCity!* playing), separated by a break of two weeks. At the beginning and at end of the experimental procedures, all patients were asked to perform a neuropsychological test, to assess their cognitive functions and impairment. The difficulty of the game was set according to patients’ condition and specific needs. At the end of the training, an ad hoc questionnaire was used to evaluate patients’ general satisfaction with *MindTheCity!* More specifically, patient’s level of engagement, intrinsic motivation and satisfaction have been evaluated through an ad hoc Patient Satisfaction Questionnaire (PSQ) administered after four weeks training as a measure of platform usability. The questionnaire is a six item, self-report questionnaire on a 10-point Likert scale, asking for: (i) how often did you (i.e., the patient) think about something else while using *MindTheCity!*, (ii) how much did you feel rewarded during the training? (iii) did you find the videogame interesting? (iv) was *MindTheCity!* fun to use? (v) did you experienced anxiety while using the videogame? (vi) was it easy to move inside the virtual environment? Overall, the questionnaire aims to investigate patient levels of distractibility (item one), gratification (item two), engagement (item three), entertainment (item four), anxiety (item five) and usability (item six). The questionnaire was administered to all patients involved (mean age: 73; SD: 5), in the study within the Department of Clinical Psychology and the Department of Geriatrics at Molinette Hospital (Turin). Notably, all patients fully complied with the protocol, carrying out the scheduled sessions until the end of the experimental procedure. Overall, there were no drop outs due to technical difficulties with the REHOME platform. Appendix A shows the list of items and response range of the administered questionnaire.

#### 2.6.2. MREP: Assessment of Motor Condition on PD Subjects

The pilot study refers to the experimental campaign organized in supervised settings, to get preliminary feedback about the system. Considering that the system has been designed to also be used in home settings, therefore with limited supervision, at this stage specific criteria have been fixed in the experimental protocol to exclude subjects with high impairment that could preclude safe and autonomous use of the system. The exclusion criteria concern severe disability assessed through the Hohen and Yahr (HY) scale (HY score > 3), severe and almost continuous tremor with inadequate response to therapy assessed through the UPDRS scale (UPDRS item score > 1), cognitive impairment assessed through the Mini–Mental State Examination scale (MMSE score < 27/30), and severe visual impairment.

A total of twenty-seven subjects were recruited for the experimental study at the Division of Neurology and Neurorehabilitation, San Giuseppe Hospital. Participants had the following overall characteristics: HY score: 2.2 ± 0.8; UPDRS motor examination total score: 31.5 ± 11.3; age: 69.8 ± 9.1 years; disease years: 8.3 ± 6.0; gender: 14 males and 13 females. There were five subjects who occasionally used the cane while walking, however, even these subjects were able to perform the motor tasks of the experimental protocol without external help.

In this experimental study, only tasks involving body movements were considered because they were more challenging in terms of user involvement and fatigue. Specifically, each participant performed the sequence of the following motor tasks twice, with a 15-min pause between the two sessions: LA, AC, PoS, Gait, LWL, FWL, and BB. The first session was used to instruct participants, thus familiarizing them with the platform. The second session was used to quantitatively measure motor performance. At the end of the second session, an ad-hoc questionnaire was administered to each participant. The questionnaire included 12 items to assess specific system features and feedback on the user experience: the proposed questionnaire was simplified with respect to standardized usability questionnaires [99], in order to explore only a subset of features relevant for the experimental context. Items were categorized to cover common aspects of usability but also specific and relevant information, including technological skills and perceived user condition. Only three alternatives were available for each item to make the choice easily understandable and more clear-cut (3-point scale), different to traditional PPSQU that are on 7-point scale. Appendix A shows the list of items of the administered questionnaire.

Upon completion of the questionnaire, the operator who supervised the patient session assessed the cooperation (i.e., whether the subject was cooperative) and comprehension (i.e., whether the subject understood the voice and text information provided by the system) using a three-point scale: this information provided an important external judgment of the user’s attitude when interfacing with the system.

Each participant was involved for less than 1 h approximately, to complete the trial and answer the questionnaire.

#### 2.6.3. MREP: Motor Rehabilitation on Subjects with Movement Disorders

The usability of the exergames was assessed through an ad-hoc experimental campaign organized in supervised settings. The experiment involved 15 subjects, recruited from the Division of Neurology and Neurorehabilitation at San Giuseppe Hospital, consisting of subjects with idiopathic PD (90%) and atypical parkinsonism. For all participants, exclusion criteria concerned cognitive disorders assessed through the MMSE scale (MMSE score < 27/30) and severe visual impairment. Again, the exclusion criteria for subjects with PD relate to severe overall disability (HY score > 3) and tremor with inadequate response to therapy (UPDRS item score > 1). Half of the recruited participants had severe motor disability, while only 30% had moderate or mild motor impairment. The sample consisted of eight males and seven females with an average age of 71.4 ± 7.2 years. Concerning patients with PD, the clinical characteristics were the following: HY score: 2.1 ± 0.9; UPDRS motor examination total score: 26.3 ± 12.5; and disease years: 6.1 ± 4.1.

At this stage, the experimental protocol established the sequence of the three exergames (CCS, PLANE, KEY) performed with a low-difficulty configuration, the same for all subjects. Each participant had to perform two sessions, with a 15-min break between them and a 5-min break between exergames to recover from fatigue. The first session was used to instruct participants and familiarize them with the system by performing multiple trials for each exergame. In the second session, a single trial was used to collect data for motor performance analysis. At the end of the second session, a traditional System Usability Scale (SUS) [100] questionnaire was administered to each participant. The questionnaire included ten standard items to assess specific features related to system usability. A total of five fixed response options (5-point scale), were available for each item, ranging from “Strongly Disagree” to “Strongly Agree”. The total score of a standard SUS questionnaire is 100 points. Appendix A shows the list of items of the administered questionnaire

Each participant was involved for approximately less than 1 h to complete the trial and answer the questionnaire.

## 3. Results

### 3.1. CRGP: Usability of Spatial Memory Domain on Elderly People with Mild Cognitive Disorders

The Cognitive Rehabilitation and Gaming Platform (CRGP) is focused on three components: (i) single domain and ecological exercises, (ii) spatial memory domain (*MindTheCity!*), and (iii) Cognitive–motor rehabilitation domain. At present, only *MindTheCity!* has already been used in a sample of healthy subjects [46] and it demonstrated to be effective in improving visuo–spatial memory and in supporting the creation of allocentric maps. *MindTheCity!* has been recently implemented in a group of patients affected by mild NCD. Currently, there are still no performance measures (i.e., number of objects recollected during the *object search* phase and angular errors during the *pointing task* phase) achievable, since the experimental procedures are still ongoing. Similarly, there are no results about the efficacy of *MindTheCity!* compared to conventional neurorehabilitation, since the control condition employed consisted of a passive stimulation, rather than a classical paper-and-pencil rehabilitation program. However, usability data based on the collection of an ad hoc PSQ are already available, thus providing interesting insights on patients’ level of engagement and intrinsic motivation and platform’s feasibility. Overall, rates of satisfaction are higher than 65% (N = 28; mean = 39; SD = 12). No differences in sex, age or education have been observed, thus demonstrating that *MindTheCity!* is suitable for use with the whole clinical population of mild NCD, without any specific excluding criteria. In addition, further analysis has been conducted in order to highlight any possible influence of other factors, such as mood alteration (depression and anxiety) and global cognitive functioning.

In order to do it, BDI-II (Beck’s Depression Inventory), STAI-Y (State-Trait Anxiety Inventory) and MoCA (Montreal Cognitive Assessment) were administered. The results obtained from the questionnaires and tests have been compared with unpaired *t*-test statistics. Specifically, subjects were divided into two independent groups based on their scores on neuropsychological tests. Subsequently, through the *t*-tests for unpaired samples, possible differences in the rate of satisfaction attributable to the scores obtained in neuropsychological tests were analyzed.

No significant correlations between the general level of satisfaction (i.e., PSQ total score) and the presence of depressive symptomatology (BDI-II total score) have been observed. However, as might be expected, PSQ item five (anxiety) was found to be significantly (t13 = 1.771, *p*-value = 0.009) influenced by the patient’s level of state anxiety (STAI-Y1 total score).

Finally, the relation between the rate of satisfaction (i.e., PSQ total score) and the global cognitive functioning (MoCA total score) has been investigated, with no significant statistical evidence obtained.

However, a significant influence (t13 = 1.771, *p*-value = 0.008) between the severity of cognitive impairment (MoCA total score) and the rate of distractibility (i.e., PSQ-item one) has been noticed. This might suggest an additional need to be supported and to motivate more affected patients during the navigational task, in order to maintain their attentional focus on it.

### 3.2. MREP: Usability of the Assessment of Motor Condition in PD Subjects

All subjects were able to perform the traditional and gamified motor tasks as indicated by the experimental protocol. The analysis of the responses collected on the PD subjects indicates that a large proportion of participants (63%) do not habitually use a personal computer (item one). Despite this, almost all participants were satisfied with the experience (item two: 96.3% of subjects rated the experience as positive).

Regarding usefulness, all subjects rated item three positively (88.9% and 11.1% of answers were Yes and Partially, respectively). A lower percentage of positive judgements concerns item four, where only 74.1% of answers were completely positive. A closer analysis reveals that those who answered No (15.9%) consider the system a complementary tool, to gym exercises and not an alternative.

Regarding usability, 81.5% of participants rated the clarity and simplicity of the system (item five) as positive (in addition, 14.8% of participants answered Partially). Moreover, 96.3% and 77.8% of participants rated the voice support (item six) and the readability of text messages (item seven) as positive, respectively. These latest results confirm the importance of adequate support to guide patients in using the system. The higher percentage to voice support suggests that it is probably more effective in giving instructions rather than text messages, which are perhaps more difficult to locate on the screen due to impaired vision in older subjects. In addition, voice support is probably more familiar to PD subjects since physicians give the same instructions during follow-up visits.

As shown in Appendix A, item eight aimed to get feedback on the difficulty of traditional evaluative motor tasks. Participants rated traditional motor tasks as Easy with the following distribution: 66.7% (LA), 81.5% (S2S), 88.9% (PoS), 48.1% (G). This result was expected, since LA and G are the most dynamic and, therefore, most difficult tasks for subjects with motor impairment. However, only a few subjects rated the motor tasks as Hard (7.4% for LA, 7.4% for S2S, 3.7% for PoS, and 14.8% for G) probably due to their higher impairment severity (HY > 2). Item nine aimed to get the same feedback on the difficulty of gamified motor tasks. Participants rated gamified motor tasks as Easy with the following distribution: 74.1% (FWL), 74.1% (LWL), 70.4% (BB). However, only a few subjects (three) rated the gamified motor tasks as Hard (11.1% for all tasks) probably due to their higher impairment severity (HY > 2). It is interesting to note that BB has a slightly higher percentage than LA (70.4% vs. 66.7%), considering that BB is a gamified version of LA. The main difference is in the lower speed of execution, since the subjects have to wait for the ball light-on before performing the leg movement. Probably this feature made execution easier for some impaired subjects. Particularly relevant in the case of gamified motor tasks is the subject involvement (item ten): almost all participants indicated a medium to high level of engagement during gamified motor tasks (85.2% for LWL and FWL; and 92.6% for BB), emphasizing the importance of realistic and enjoyable game scenarios, especially from the perspective of prolonged rehabilitative treatment.

Finally, none of the participants reported feeling particularly fatigued (item eleven) at the end of the session (No: 48.1%, Partially: 51.9%, Yes: 0.00%). In addition, 96.3% of participants rated the overall experience positively (item twelve). Based on the items for the supervisor, it appears that all participants were cooperative (Yes: 85.2%, Partially: 14.8%, No: 0.0%), and almost all understood the instructions provided by the system through audio and text messages on the screen (Yes: 70.4%, Partially: 29.6%, No: 0.0%), without needing clarification from the supervisor.

The results indicate an overall positive feedback from participants, who were able to use the system through the user interface and complete the session of traditional and gamified tasks. This confirms the usability and acceptability of this component of the MREP platform, which could also be used effectively in minimally supervised settings for the assessment of motor performance in pathological subjects. Table 5 shows a final summary of positive judgement rates grouped by questionnaire categories.

### 3.3. MREP: Usability of Motor Rehabilitation on Subjects with Movement Disorders

All participants were able to perform the exergames as indicated by the experimental protocol. Analysis of the SUS score on all participants confirms the overall usability of the exergames: in fact, an average value of 69.0 ± 22.1 points was obtained (68 points is minimum value for a system to be considered usable). This is particularly relevant in the case of pathological subjects, that commonly have more difficulties in using technological solutions. In more detail, 70% of participants confirm they would like to use the exergames frequently (item one), while 20% would not. This suggests that the subjects had high engagement and enjoyment of exergames in the virtual environment. Only 10% of the sample found the exergames unnecessarily complex (item two) and not easy to use (item three), while nearly 90% rated the complexity positively and more than 50% of participants rated ease of use positively.

Regarding item four, some problems emerged, as only 30% of participants believed that they would not need the support of a technician to use the system, while another 30% would agree to receive some kind of technical support. The remaining 40% are neutral on this point. This suggests that the introduction of additional support (e.g., voice support) could be helpful in overcoming this problem. On the other hand, 70% and 100% of participants found that the various functions of the applications are well integrated (item five) and consistent (item six). Concerning item seven, 60% of participants believe that most people would learn to use this system very quickly, while only 30% do not believe so.

In addition, 60% of the sample found the exergames quite difficult in general (item eight). This is probably a consequence of the degree of motor impairment of many participants, which would have required a finer and proper adjustment of the difficulty level of each exergame. The adjustment of exergame difficulty according to motor condition and its evolution is one of the goals of the ongoing mid-term experimentation, as envisaged by the project. However, more than 60% of the participants felt very confident in using them (item nine), probably due to performing the exergames in a sitting position. However, the inconsistency between items eight and nine could also be explained by the last item (item ten) which shows that 60% of the participants had to learn a lot of information before they could start using exergames. In fact, after initial difficulties (first session), they learned to use the system without any problems, even though it was the first time the participants used the exergames. This result confirms the need for good initial training, but more importantly it suggests that continuous practice helps to master the system more and more, which is crucial in the case of rehabilitation protocols over long periods. Table 6 shows a final summary of positive judgement rates grouped by questionnaire categories.

## 4. Discussion

Demographic changes associated with an aging population pose new challenges and raise important questions about the design of appropriate health care sustainability strategies for the near future [11]. Indeed, as recently demonstrated, aging is correlated with an increase in chronic and disabling pathologies that require prolonged periods of treatment to ensure a satisfactory quality of life. Among these, neurological and neurodegenerative diseases or consequences of acute stroke events are the main causes of severe disability in the elderly, impacting cognitive and motor domains. In this scenario, rehabilitation programs are helpful in mitigating the adverse effects and ensuring improvement in quality of life, especially in progressive and chronic conditions [7,8,9]. Some studies have also shown that greater improvements were achieved by ensuring continuity of care in the home environment [10]. An enhancement of health services could take place using new ICT solutions, where technologies could provide valuable support for both monitoring and rehabilitation purposes in the home and minimally supervised settings [13,14]. Currently however, numerous barriers prevent widespread use of such solutions in elderly and pathological individuals, including aspects of usability, acceptability, motivation, skepticism, and comfort in using digital tools [10,15,16].

The REHOME solution aims to overcome some of the common weaknesses that hinder the deployment of telemedicine platforms and e-health solutions for remote monitoring and rehabilitation.

In fact, compared to the state-of-the-art [12,13,14,18], the REHOME solution aims to cover some of the reported limitations through the following innovative aspects:(a)The development and integration of various sensors and methodologies for multi-source signal collection, thus facilitating a more comprehensive analysis of patients’ overall condition and treatment effects. This novelty intends to overcome the limitations of many eHealth platforms and services that focus only on specific individual aspects without providing a comprehensive overview of the patient’s condition. Instead, this novelty could be particularly relevant from a clinical perspective, especially in complex and multi-system pathologies such as those considered by the REHOME project;(b)The use of recent, innovative, and promising methodologies in the context of motor and cognitive rehabilitation, such as gamification and exergaming in virtual environments, to favor patient condition assessment and motor–cognitive training in playful, fun, life-situation-inspired, and rewarding environments, with a focus on patients’ engagement, needs, motivations, and aesthetic preferences [101,102]. This novelty intends to foster greater patient involvement and satisfaction, thus enticing the patient to continue with the treatment according to the therapeutic plan. In this way, it could be possible to overcome the decline in interest that is one of the main issues of technological solutions as emerges in the literature;(c)Personalization of rehabilitation treatment and monitoring of different domains (motor, cognitive, sleep) through the definition of a customizable plan (e.g., in terms of type, difficulty, frequency, and duration of exercises) based on the patients’ needs, health conditions, and progress in achievement of new rehabilitation goals. This feature allows the patient’s current condition to be taken into account. On the one hand, this avoids discouraging him or her with exercises that are too difficult or stressful. This allows the therapist to easily set new therapeutic goals (e.g., increasing the difficulty or duration of exercises) that can be achieved by the patient. The lack of customization is another of the weaknesses highlighted in the literature, which does not allow a technological solution to easily and quickly adapt to different scenarios and specific needs;(d)High usability and interaction, through user interfaces specifically designed to facilitate the use of individual platforms, even in the home and unsupervised scenarios. This aspect allows the solution to be easily usable by the patient or with the support of a caregiver, but without specific technological expertise. The complexity, difficulty of use, and poor comfort of sensors and devices are known issues in the literature, which cause technological solutions to be gradually abandoned because they are often considered too complex and impractical;(e)The development of a scalable, flexible, easily extendible, cloud-based, and distributed microservices architecture based on standard HL7 FHIR protocol and models. This architectural style aims to reduce the costs of integrating additional or third-party home platforms and services to monitor and rehabilitate other pathological conditions. Moreover, using HL7 FHIR improves interoperability and data exchange with other healthcare infrastructures.

This was achieved through close collaboration between health care professionals and technicians, which resulted in meeting the needs of patients (and clinicians) through the design and requirements of a comprehensive solution. In fact, clinicians, researchers, and technicians selected the most appropriate technological and methodological approaches to maximize patients’ compliance, technical reliability, data collection, and, to provide the necessary tools to ensure continuity of follow-up and rehabilitation treatment remotely, thus not only in traditional inpatient and outpatient settings.

The experimental phase of the REHOME project, which ultimately aims to evaluate the effectiveness of this new integrated treatment approach, is currently underway. However, some pilot trials on smaller samples of target subjects, have already been completed with the goal of getting initial feedback related to aspects of usability and patient satisfaction. Specifically, three experimental trials involving different groups of pathological subjects were organized to investigate the peculiar usability features of some components of the REHOME solution with respect to the target population.

The first experimental study focused on the spatial memory domain component of the CRGP platform and involved 28 elderly subjects with mild neurocognitive disorder (DMS-5). An ad-hoc questionnaire was designed to investigate the peculiarities of usability and patient’s satisfaction of a 3D exergame in which a gamified systematic approach and aesthetically pleasing backgrounds were adopted, as several studies have shown that these two elements are able to support patients’ motivation and learning outcomes [46,101,103,104]. The study also included neuropsychological tests, online behavioral assessments, and specific in-person electrophysiological protocols to monitor the rehabilitation course of patients [103,105,106,107]. Preliminary results of the questionnaire showed an overall satisfaction rate of more than 65%, with no significant modulations induced by the presence of depressive symptomatology (BDI-II total score) and global cognitive functioning (MoCA total score). However, further analysis of individual items showed the strong influence of depressive symptoms and of anxiety perception (item five of the questionnaire), indicating that more depressed subjects were likely to experience anxiety using the 3D exergame. Moreover, a strong relationship was observed between the severity of cognitive impairment and distractibility (item one of the questionnaire), that the most affected patients may need to be supported and motivated to maintain attention on the exergame.

The second experimental study focused on the MREP platform and, in particular, the assessment of motor condition through traditional and gamified motor tasks [73,76,79,102]. The study involved 27 subjects with Parkinson’s disease. An ad-hoc questionnaire was designed to investigate the peculiarities of usability and patient’s satisfaction of this component in which optical approaches and body tracking algorithms were used to implement a non-invasive motion capture system aimed at simplifying human-computer interaction, even in virtual environments. Preliminary results of the questionnaire showed an overall satisfaction rate (item two of the questionnaire) of more than 96%, despite the fact that a large percentage of participants (63%) do not routinely use a personal computer (item one). Similarly, a high percentage of participants rated positively (more than 87%) the aspects of usability, usefulness, ease of use, and engagement. In addition, no participant reported feeling particularly fatigued at the end of the test session: this is a relevant finding, especially from the perspective of continued and prolonged use over time. Finally, particularly interesting are the results of the supervisor items, an external judgement that confirms the cooperative attitude of the participants and the comprehensibility of the instructions provided by the system through the message and audio support.

The last experimental study focused on the MREP platform and, in particular, the motor rehabilitation component through three exergames in the virtual environment. The study involved 15 subjects with Parkinson’s disease and other movement disorders. In this case, a traditional SUS questionnaire was used. Preliminary results confirm an average usability score above 68 points, which is the minimum threshold for considering a system usable. An in-depth analysis of individual items revealed specific valuable information. More than 70% of participants said they would be willing to use exergames frequently (item one of the questionnaire): this suggests a higher user satisfaction in using exergames. Regarding ease-of-use, participants rated item two and item three very positively, while worse results were obtained for item four (technical support to use the system). However, this result may have been influenced by the fact that this was the first-time participants were using exergames (item eight); continued practice and learning over time should foster progressive independence in the use of exergames (as results from item seven and item ten), allowing this critical issue to be improved. Another positive judgement was related to coherence (item five and item six), and confidence in using the exergames (item nine). However, the overall result of the questionnaire is positive, especially considering the severity of the people involved in the experimental study. These results on usability of the MREP components using RGB-Depth sensors confirm the suitability of vision-based approaches for designing innovative rehabilitation strategies to support traditional physiotherapy and exercises [108,109,110,111,112].

Preliminary results on usability and the platform’s ability to train and characterize different functional aspects are certainly promising. The final experimental phase currently underway will allow us to confirm the results obtained and presented on a larger number of participants, but more importantly, to evaluate the effectiveness of the innovative and integrated treatments provided by the REHOME solution. For the ongoing final step, dedicated experimental protocols have been defined for motor and cognitive rehabilitation, with a control group subjected only to a conventional rehabilitation protocol and a target group also subjected to a gamified rehabilitation protocol. The goal is to detect differences in performance between the groups at the end of the experimental phase: this will allow us to present findings on the effectiveness, potential, and limitations of gamified rehabilitation treatments in support of conventional rehabilitation treatments. In addition, the final experimental phase will also allow us a more detailed and comprehensive overall evaluation of the solution. The questionnaires administered for evaluating patient-side usability will be complemented by interviews with HCPs aimed at assessing other aspects, including statistics on the time spent by HCPs for the follow-up of each patient, critical issues encountered in staff management, statistics on technical problems, the level of satisfaction of HCPs in using the solution, and patients’ adherence to the scheduled sessions of the personalized treatment plan. Of course, final experimentation will allow us to also improve the usability analysis by including the other components that were not explicitly considered in this work and to take advantage of the potential benefits offered by the REHOME solution.

From a research perspective, the integrated design of the REHOME solution could indeed provide additional insights to identify the most sensitive markers to characterize patient performance and changes in the pathological condition. From a technological and clinical point of view, some technical features of the REHOME solution (including modularity, extensibility, and customization) mean that it can be quickly (and easily) adapted and extended to meet other new future and specific needs of patients, caregivers, and healthcare professionals. From the perspective of patients, the REHOME solution offers the possibility of being monitored in a home and a more familiar environment, increasing the frequency of assessments, and favoring continuity of treatments. All this while remaining in contact with clinical staff without traveling to healthcare facilities, which is often a burden for frail patients and families. From the healthcare system’s perspective, solutions such as the one developed in REHOME could enable, in the long term, a decrease in the cost of healthcare services (e.g., through optimizing treatments based on the patient’s actual condition, reducing hospitalizations for the activation of new rehabilitation programs, and optimizing healthcare resources), and ensure the future sustainability and accessibility of healthcare services by including remote technological solutions as new strategies to support clinical management of patients.

In contrast, the main limitations are as follows: the developed solution requires an internet connection to enable remote patient follow-up, preventing its use in areas not covered by robust and reliable connections (this is a frequent limitation in this type of solution); the computational resources required to ensure real-time motion tracking (with deep learning techniques) are still high, preventing the deployment of this component on a large scale; the solution is not suitable for individuals with disabilities that are too severe for whom a supervised environment is still preferable; good initial training is necessary to use each component of the solution correctly and independently by patients and caregivers.

The solution developed within the REHOME project thus belongs to the category of ICT tools for e-health, meeting the needs for innovative and highly flexible technological solutions for rehabilitation and fulfilling the guidelines of Horizon Europe and the Italian PNRR (https://italiadomani.gov.it/it/home.html accessed on 25 October 2022).

## 5. Conclusions

The paper describes the solution developed as part of the REHOME project, which integrates innovative technologies and methodologies aimed at remote rehabilitation (and monitoring) of cognitive, motor, and sleep disorders related to neurological diseases. The literature review highlighted the main obstacles and problems yet to be solved to ensure a deployment of ICT solutions to support the health service. Usability and acceptability assessment is one of the main obstacles. After a comprehensive overview of the implemented solution, the paper focused on some specific components that were considered for a preliminary usability evaluation. The questionnaires show that, in general, participants in the experimental trials rated their experience positively, which is a relevant result since it was achieved on elderly and pathological subjects. This makes us confident that the implemented solution could be effectively used in the home setting and could ensure the continuity of rehabilitation protocols and improvement of patients’ quality of life.

## Figures and Tables

**Figure 2 sensors-22-09467-f002:**
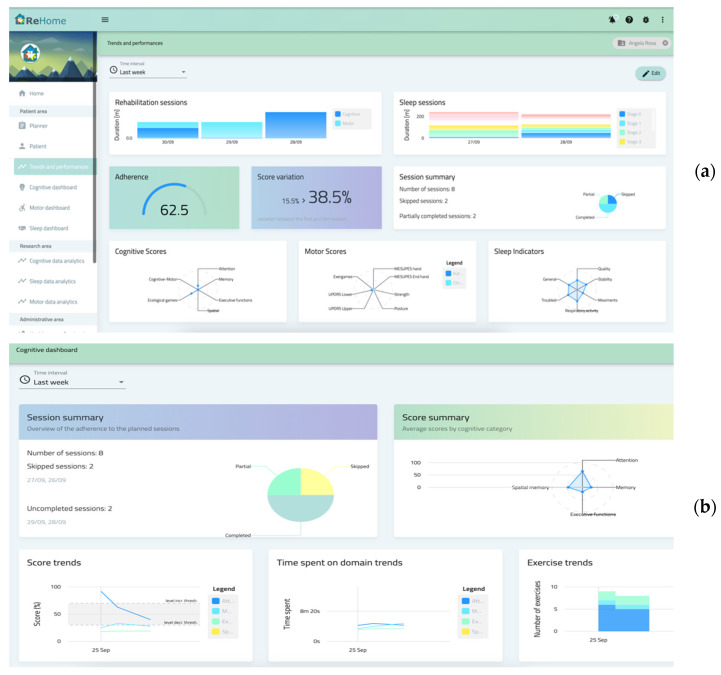
Graphical user interface for HCPs: generic (**a**), cognitive (**b**), motor and electromyography (**c**), sleep (**d**) monitoring dashboards.

**Figure 3 sensors-22-09467-f003:**
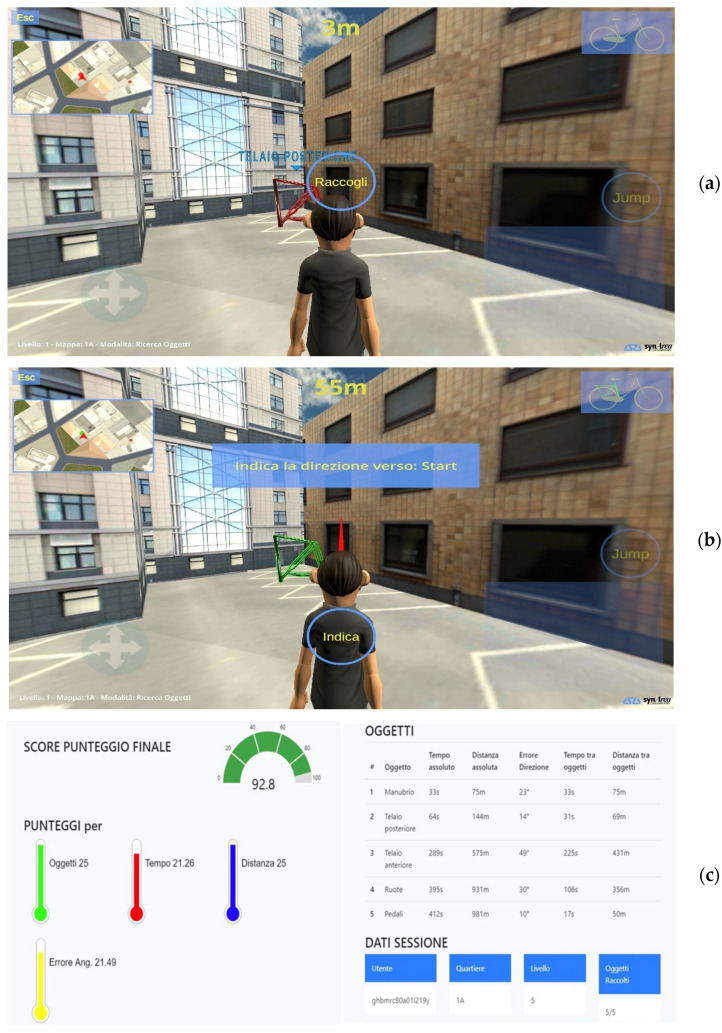
*MindTheCity!* screenshots for object search task (**a**), pointing task (**b**), and results (**c**).

**Figure 4 sensors-22-09467-f004:**
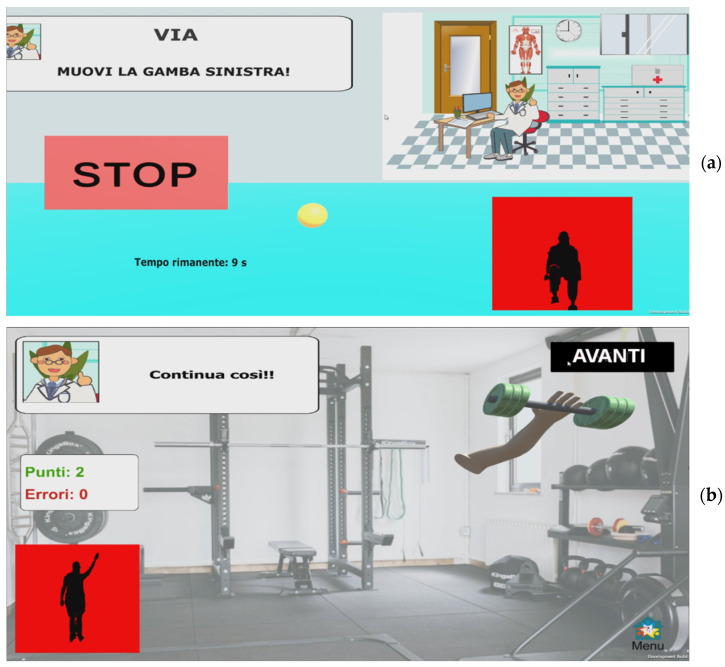
Example of a GUI for a LA task (**a**), FWL exergame (**b**), and upper limb tasks (**c**).

**Figure 5 sensors-22-09467-f005:**
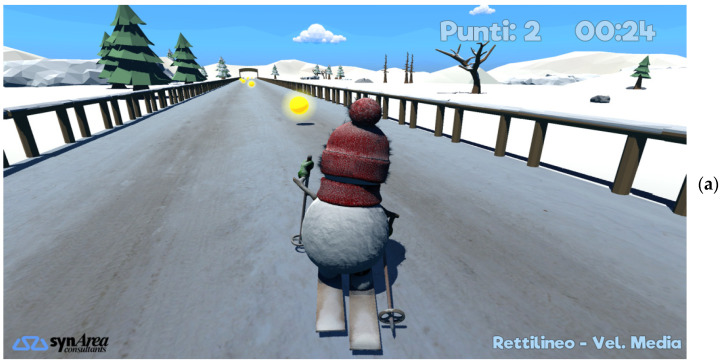
Cross-country skiing (**a**), Airplane (**b**), and Keyboard (**c**) exergames.

**Table 1 sensors-22-09467-t001:** Summary of the main functionalities of the CRGP.

Domain	Skills	Approach	Technology/Devices
Spatial memory domain	Visuo-spatial	3D Videogame spatial navigation in a virtual city (*MindTheCity!*)	Tablet, keyboard or joystick
Cognitive–motor rehabilitation domain	Topographical orientation Selective attention Object categorization Motor planning Learning and recall of visual information Motor inhibition	3D Videogame—exploring a virtual desert island to activate specific mini-games (*MindTheCraft!*)	Tablet, keyboard or joystick
Single domain and ecological exercises	Attention Executive functions Long-term memory Learning	360-degree exploration of environments and cognitive exercises in daily life virtual contexts	Tablet

**Table 2 sensors-22-09467-t002:** Summary of the main functionalities of the MREP.

Function	Skills	Approach	Technology/Devices
Assessment of motor condition	Mobility of upper and lower limbs Motor coordination Postural attitude Postural instability Balance Gait Hand dexterity	Standardized motor tasks and exercises in virtual environment using non-invasive body tracking algorithms	RGB-Depth camera (Azure Kinect DK, Microsoft^®,^ Microsoft Corporation, Redmond, WA, USA)
Motor rehabilitation	Motor control and coordination of upper limbs Trunk flexion Arm extension and pointing	3D exergames in virtual environment	RGB-Depth camera (Azure Kinect DK, Microsoft^®^, Microsoft Corporation, Redmond, WA, USA)
Upper limb rehabilitation	Hand motor dysfunctions and dexterity (finger extension, cylindrical and spherical grasp)	Hand movement exercises using sEMG	sEMG Armband (REMO^®^, Morecognition s.r.l., Turin, Italy)

**Table 3 sensors-22-09467-t003:** Summary of the main signals monitored by SEP.

Function	Signals and Parameters	Technology/Devices
Evaluation of sleep disorders	Respiratory rate, hearth rate (ECG)	Chest strap (sensor prototype by Astel s.r.l., Pavone Canavese, Turin, Italy)
	EEG, EOG, orinasal flux	Cap (sensor prototype by Astel s.r.l., Pavone Canavese, Turin, Italy)
	Periodic Leg Movements	Socks (sensor prototype by Astel s.r.l., Pavone Canavese, Turin, Italy)
	Room noise, temperature, humidity, illumination	Environmental sensor (Omron^®^, Kyoto, Japan)
	Presence in bed, quantity of movement, respiration and hearth rates	Pressure band (Emfit QS^®^, Emfit Ltd., Vaajakoski, Finland)
	Relevant body movement events	RGB-Depth camera (Azure Kinect DK, Microsoft^®^, Microsoft Corporation, Redmond, San Francisco, CA, USA)—only infrared streaming

**Table 4 sensors-22-09467-t004:** Characteristics of the experimental trials.

Platform	# Subjects	Age	Target	Exclusion Criteria
CRGP—Spatial memory domain	28 (14 M/14 F)	73.0 ± 5.0	Elderly people affected by mild neurocognitive disorder (DSM-5)	<55 years old <5 years education Severe loss of autonomy Sensory deficit that prevents tablet’s use
MREP—Assessment of motor condition	27 (14 M/13 F)	69.8 ± 9.1	Subjects affected by Parkinson’s disease	MMSE score < 27/30 Severe visual impairment HY score > 3 Severe tremor
MREP—Motor rehabilitation	15 (8 M/7 F)	71.4 ± 7.2	Subjects affected by Parkinson’s disease, atypical parkinsonism	MMSE score < 27/30 Severe visual impairment HY score > 3 Severe tremor

**Table 5 sensors-22-09467-t005:** Summary of positive rating percentages on the categories of the ad-hoc questionnaire.

Questionnaire Categories (Reference Items)	% Positive Rating ^1^ (N = 27)
Usefulness (items 3,4) ^2^	87.1%
Usability (items 5,6,7) ^2^	90.1%
Easy-of-use (item 8)	91.7%
Easy-of-use (item 9) ^3^	88.9%
User Engagement (item 10) ^3^	87.7%
User Perceived Status (items 11,12) ^2^	98.2%
Overall Satisfaction (item 2)	96.3%

^1^ Zero-point responses were considered as a negative rating (Appendix A). ^2^ Average percentage of questionnaire items shown in parentheses. ^3^ Average percentage on traditional and gamified motor tasks.

**Table 6 sensors-22-09467-t006:** Summary of positive rating percentages on the categories of the SUS questionnaire.

Questionnaire Categories (Reference Items)	% Positive Rating ^1^ (N = 15)
User satisfaction (item 1)	70.0%
Easy-of-use (items 2, 3, 4) ^2^	53.0%
System coherence (items 5, 6) ^2^	85.0%
Usability (items 7, 8, 9, 10) ^2^	60.0%
Overall Satisfaction ^3^	50.0%

^1^ Neutral and negative responses (according to specific items) were considered as a negative rating (Appendix A). ^2^ Average percentage of questionnaire items shown in parentheses. ^3^ Percentage of overall SUS scores above the system usability threshold.

## Data Availability

Data are available on request.

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
