# Peer review of "Usability of the REHOME Solution for the Telerehabilitation in Neurological Diseases: Preliminary Results on Motor and Cognitive Platforms"

_sensors, 2022, doi:10.3390/s22239467_

Round 1

Reviewer 1 Report

This paper presents the background of the technological solution for the REHOME project. The project integrates methodologies and devices for remote monitoring and rehabilitation of cognitive, motor, and sleep disorders associated with neurological diseases. The paper shows the preliminary results of usability and user experience questionnaires on 70 subjects recruited in three experimental trials. The paper is well organized and clearly written and the methodology is correctly described too.

It is stated that this paper is an extended and more detailed version of the recently conference paper presented at the second IEEE Conference on ICT solutions for eHealth (ICTS4eHealth 155 2022). Please define clearly the main differences between both papers since I did not have access to it.

Regarding the architecture of the solution for the REHOME system, it has been developed with cloud technologies and is based on a distributed microservices architecture, what seems very appropriate for scaling the system and include new microservices as long as the project grows.

Regarding interoperability and standards, and security aspects, it has been considered and guaranteed by an API Gateway based on the HL7 FHIR R4 standard and LOINC for the exchange of health data, which supports RESTful communication over HTTPs protocol, OpenID Connect security standards and encrypted channel. Security requirements are met to make the platform compliant with Regulation (EU) no. 2016/679 (GDPR). It has been considered to ensure interoperability in the proposal in all aspects of the project.

The only concern with the methodology is that more deep details about the architecture should be provided: the paper goes from a high level REHOME system architecture (Fig. 1) to the GUI for HCPs, etc. (Figs. 2, 3, 4 and 5) without an intermediate description of the technology core.

The evaluation of the project presented is mainly based on the preliminary results of usability and user experience obtained from patient questionnaires. However, one would expect (I guess it is ongoing or future work) a more detailed and complete evaluation study: a complete evaluation plan including more indicators regarding e.g., staff time dedicated to every patient, staff organization, technical problems found, supervisor staff satisfaction, economical aspects, etc. indeed a complete viability and impact study of the new services provided from several perspectives. There are some frameworks to evaluate eHealth services that can be considered to improve the surveys and add more information to the questionnaires used.

Finally, the following minor comments/questions raise to my mind:

Is there any comparison between the REHOME tele approach vs. the conventional rehabilitation in the patient’s study group?  Where the patients involved in the trials previously involved in a classical rehabilitation treatment?

How long did the patients participate in the study? Adherence to these systems often declines over time...  Some type of chronogram or time information would be helpful.

Which would be the economic cost of the system for a patient/ insurance carrier to use it for every different functionality of the MREP? I guess patients used a prototype in the lab, but how it can be transferred to patient’s home? At which cost?

Perhaps more emphasis could be placed on the novelty of this proposal compared to other…

Reviewer 2 Report

This study aimed to propose a solution for telerehabilitation in neurological diseases. I have the following suggestions.

What is the novelty of this study although several solutions for telerehabilitation in neurological diseases have been proposed earlier?

Please add a paragraph about the contribution of this article in a bulleted form at the end part of the Introduction section.

Authors should improve conceptual figures of their proposed subsystems with more details and parametrization.

The manuscript is difficult to follow, there are so many systems and case studies with lack of details.

Experimental methodology should be detailed for each case study.

Authors need to add a table including the data structure, details of data source, and size of dataset.

Data processing should be more detailed and signal specific.

Authors may explore studies related to ML application the sleep stage prediction. As example, EEG are investigated for Sleep stage prediction in article, quantitative evaluation of eeg-biomarkers for prediction of sleep stages.

Authors may introduce the remote patient monitoring applications in broader scope, such as article, healthsos: real-time health monitoring system for stroke prognostics; and in article, big-ecg: cardiographic predictive cyber-physical system for stroke management.

Authors should report objective performance measures of their systems. Only subjective performances were reported.

Authors must have discussion on the advantages and drawbacks of their proposed method with other studies adding a table in discussion section.

Round 2

Reviewer 1 Report

The new version of the article has been successfully improved. The authors have adequately responded to all the questions raised, and the changes made to the article have served to improve the aspects that required some clarification. Congrats for the work done.

Reviewer 2 Report

Visualization of results should be improved. 
